# Mdivi-1 Induced Mitochondrial Fusion as a Potential Mechanism to Enhance Stress Tolerance in Wheat

**DOI:** 10.3390/life12091386

**Published:** 2022-09-06

**Authors:** Daniya Rakhmatullina, Anastasia Mazina, Anastasia Ponomareva, Svetlana Dmitrieva, Richard Peter Beckett, Farida Minibayeva

**Affiliations:** 1Kazan Institute of Biochemistry and Biophysics, FRC Kazan Scientific Center, Russian Academy of Sciences, 420111 Kazan, Russia; 2School of Life Sciences, University of KwaZulu-Natal, Scottsville 3209, South Africa; 3Institute of Fundamental Medicine and Biology, Kazan (Volga Region) Federal University, 420008 Kazan, Russia

**Keywords:** wheat, mitochondria, fusion, wounding, energy status, redox metabolism

## Abstract

**Simple Summary:**

Mitochondria play a key role in providing energy to cells. This paper is dedicated to elucidating mitochondria-dependent mechanisms that may enhance abiotic stress tolerance in wheat. Mitochondria are constantly undergoing dynamic processes of fusion and fission. In plants, stressful conditions tend to favor mitochondrial fusion processes. The role of mitochondrial fusion was studied by applying Mdivi-1, an inhibitor of mitochondrial fission, to wheat roots subjected to a wounding stress. Increased mitochondrial functional activity and upregulation of genes involved in energy metabolism suggest that mitochondrial fusion is associated with a general activation of energy metabolism. Controlling mitochondrial fusion rates could change the physiology of wheat plants by altering the energy status of the cell and helping to reduce the effects of stress.

**Abstract:**

Mitochondria play a key role in providing energy to cells. These organelles are constantly undergoing dynamic processes of fusion and fission that change in stressful conditions. The role of mitochondrial fusion in wheat root cells was studied using Mdivi-1, an inhibitor of the mitochondrial fragmentation protein Drp1. The effect of the inhibitor was studied on mitochondrial dynamics in the roots of wheat seedlings subjected to a wounding stress, simulated by excision. Treatment of the stressed roots with the inhibitor increased the size of the mitochondria, enhanced their functional activity, and elevated their membrane potentials. Mitochondrial fusion was accompanied by a decrease in ROS formation and associated cell damage. Exposure to Mdivi-1 also upregulated genes encoding the glycolytic enzyme glyceraldehyde-3-phosphate dehydrogenase (GAPDH) and an energy sensor AMP-dependent protein sucrose non-fermenting-related kinase (SnRK1), suggesting that mitochondrial fusion is associated with a general activation of energy metabolism. Controlling mitochondrial fusion rates could change the physiology of wheat plants by altering the energy status of the cell and helping to mitigate the effects of stress.

## 1. Introduction

In eukaryotic cells, mitochondria perform many functions, including the production of ATP during oxidative phosphorylation, maintenance of cytosolic calcium levels, and the generation of reactive oxygen and nitrogen species [1]. Mitochondria are dynamic organelles that undergo fission and fusion; they can change their morphology, intracellular localization, and number. The size and number of mitochondria is mainly determined by the balance between their fusion and fission [2,3]. This balance corresponds to changes in environmental conditions and aims to optimize cellular metabolism [4]. A shift towards fusion leads to the formation of interconnected mitochondrial networks, while a shift towards fission induces the formation of more numerous morphologically and functionally different small spherical organelles [5,6]. In animals, it has been shown that such discrete mitochondria may become dysfunctional [1]. Furthermore, fusion of mitochondria can be both transient, when the morphology and composition of mitochondria do not change, or more prolonged, providing a dynamic network where the contents of the matrix and membrane proteins and lipids can be exchanged [7], maintaining the quality of mitochondria [8,9].

It has been demonstrated that the frequencies of fusion and fission events are closely related to the energy status of mitochondria [9,10]. ATP synthesis is carried out by an ATP synthase (complex V—oxidative phosphorylation systems) following the generation of a proton-motive force by protein complexes I–IV in the inner mitochondrial membrane. Studies using inhibitors of protein complexes [11,12,13] and uncouplers [14] have shown that a decrease in the activity of these complexes can cause changes in mitochondrial morphology. Many stresses are known to decrease the activity of the complexes, and therefore change mitochondrial morphology. For example, in mammals, interconnected mitochondrial networks are frequently present in cells with active respiration and metabolism, while fragmentation of mitochondria often occurs in response to stresses [15]. In plants, the formation of giant mitochondria occurs at critical times of the life cycle, and a variety of stresses and mitochondrial inhibitors have been reported to have a variety of effects on mitochondrial morphology [9]. Giant mitochondria are formed due to hypoxia or in response to reduced cytosolic sugar levels [16]. Recently, the formation of giant mitochondria was observed in etiolated cotyledons of *Arabidopsis thaliana* seedlings germinated in the dark, and their structure studied by making serial TEM sections [17]. The appearance of giant mitochondria occurs in response to the application of respiratory inhibitors. For example, [18] showed that elongated mitochondria were formed by treating *Nitella flexilis* cells with the mitochondrial poisons antimycin A and cyanide, and the inhibitor of electron transport in the chloroplast 3-(3,4-dichlorophenyl)-l,l-dimethylurea (DCMU) also induced the formation of elongated mitochondria. This was accompanied by a decrease in the activity of mitochondrial complexes III and IV and in photosynthetic activity. In the roots of wheat seedlings, inhibitors of various mitochondrial ETC complexes, e.g., rotenone and antimycin A, induced the temporary formation of giant mitochondria [19]. In contrast, cyanide and the uncouplers of oxidation and phosphorylation 2,4-dinitrophenol and carbonylcyanide-l-chlorophenylhydrazone (CCCP) had no effect on the size and shape of mitochondria in tobacco cultured cells [20]. Therefore, in plants the relationship between the activity of the mitochondrial complexes and mitochondrial architecture remains uncertain.

The control of mitochondrial fusion and fission has been evolutionarily conserved, and is very similar in all organisms, from yeast to humans [6]. The architecture and dynamic reorganization of the mitochondrial network (“mitochondrial dynamics”) are regulated by several recently identified proteins, most of which are dynamin-like proteins. These proteins belong to the GTPase families. For fusion, in tobacco the GTPase Miro2 in mitochondria interacts with the ER and appears to control mitochondria fusion and motility [21]. Conversely, the dynamin related protein (Drp1) participates in mitochondrial fragmentation. In mammalian cells, inhibition of Drp1 leads to elongation of mitochondria and delays their fragmentation during apoptosis [22]. Mitochondrial fusion is regulated by mitofusins (Mfn1, Mfn2), which regulate the fusion of the outer membrane, and the optical atrophy protein (Opa1), which promotes the fusion of the inner membrane. Collectively, these proteins play an important role in mitochondrial biogenesis, regulation of the cell cycle, cell survival, pathology, and death [23,24,25].

In *Arabidopsis*, homologues of the fragmentation protein DRP1 (AtDRP3A and AtDRP3B) have been identified [26]. These proteins are cytosolic and bind to the mitochondrial fragmentation site using the Elongated Mitochondria 1 (ELM1) protein, which is on the outer mitochondrial membrane [27]. For fusion, the precise mechanisms controlling the process in plants remain unclear [28]. While close homologues of the fusion proteins of the outer mitochondrial membrane are found in plants, they are localized in chloroplasts and determine thylakoid architecture [29]. Some results suggest that increased expression of the gene encoding the MELL1 protein, which controls the interaction of mitochondria and ER in plant cells, can lead to an increase in the size of mitochondria. However, this protein does not contain a GTPase domain, and the precise role of the protein in fusion is unknown [30].

Mitochondrial division inhibitor 3-(2,4-dichloro-5-methoxyphenyl)-2,3-dihydro-2-thioxo-4 (1H)-quinazolinone (Mdivi-1), an inhibitor of dynamin GTPases, has been widely used in animal cells to pharmacologically prevent mitochondrial fragmentation, e.g., in response to stress [31]. Mdivi-1 was first described by Cassidy-Stone et al., [32] as an inhibitor of Drp1 activity in yeast and mammalian cells. The authors demonstrated that Mdivi-1 prevents the oligomerization of Drp1 molecules into a ring, which occurs during mitochondrial fragmentation. Mdivi-1 promotes the rapid formation of a mitochondrial network [1] and prevents the degradation of mitochondria during autophagy [33]. Treating a variety of cells with this compound reduces the formation of reactive oxygen species (ROS) and inhibits apoptosis [34,35]. In plants, we have shown that treating wheat roots with Mdivi-1 induces the formation of “megamitochondria” [19]. The aim of the present work was to study the morphological and physiological consequences of mitochondrial fusion in plant cells in order to increase our understanding of the role of mitochondrial dynamics in the response of plants to stresses. Initially, the ultrastructure of megamitochondria was studied. Mitochondrial energy status and the expression of genes encoding the glycolytic enzyme glyceraldehyde-3-phosphate dehydrogenase (GAPDH) and the key energy sensor AMP-dependent protein sucrose non-fermenting-related kinase (SnRK1) were analyzed. Results suggested that mitochondrial fusion is accompanied with the redistribution of energy-producing metabolic pathways to sustain the supply of substrates to fused mitochondria. Furthermore, it was found that mitochondrial fusion lowers the level of oxidative stress and autophagy.

## 2. Materials and Methods

### 2.1. Materials

Seeds of wheat (*Triticum aestivum* L., cv. Kazanskaya Yubilejnaya) were distributed evenly in a single layer and sown on a tissue wetted with distilled water on a sheet of glass for 24 h. After that imbibed seeds were transferred to the containers with 0.25 mM CaCl_2_ solution and seedlings were grown hydroponically for 5 d at a light intensity of 100 W m^−2^ 25 °C, with a 12 h photoperiod. Roots were excised from seedlings and incubated in 0.25 mM CaCl_2_ (control) and 0.1 μM Mdivi-1 (3-(2,4-dichloro-5-methoxyphenyl)-2,3-dihydro-2-thioxo-4(1H)-quinazolinone) (Sigma-Aldrich, Shanghai, China) for 1 to 5 h. Mdivi-1 was first dissolved in dimethylsulfoxide (DMSO, 1 mg in 100 μL), then brought to the working concentration with a solution of 0.25 mM CaCl_2_. A corresponding amount of DMSO was added to the control.

### 2.2. Confocal and Electron Microscopy

Longitudinal sections of root tips in the extension zone (0.5 cm) were stained with the fluorescent dyes tetramethyl rhodamine methyl ester (TMRM, Sigma, λ_ab_ 540 nm/λ_em_ 573 nm) to visualize mitochondria and assess the membrane potential and with LysoTracker Red DND 99 (LT, Invitrogen, λ_ab_ 577 nm/λ_em_ 590 nm) to visualize autophagosomes. Root sections were incubated in a solution containing 1 µM digitonin and 1 µM dye for 15 min, then rinsed three times with a solution containing 20 mM HEPES-Tris pH 7.4, 0.25 M sucrose, 5 mM EDTA, 0.1% BSA, 1 mM MgCl_2_. Fluorescence intensity was recorded using an LCM 510 META confocal microscope (Zeiss, Göttingen, Germany).

To study ultrastructure, 1–2 mm long root segments from the elongation zone were fixed with 2.5% glutaraldehyde solution in phosphate buffer pH 7.2 for 2 h. Samples were then dehydrated in ethyl alcohol of ascending concentration (30°, 40°, 50°, 60°, 70°, 96°), then acetone and propylene oxide. The samples were polymerized for 3 d in Epon-812 (Serva, Heidelberg, Germany) at temperatures from 37 °C to 60 °C. Sections were cut using an LKB-III ultramicrotome (Sweden) and stained with a saturated solution of aqueous uranyl acetate for 10 min at 60 °C, and then for 10 min with an aqueous solution of lead citrate. The preparations were examined using transmission electron microscopy (Jem-1200 EX, Jeol, Tokyo, Japan).

### 2.3. Respiration Rates

The oxygen consumption by roots was recorded by the manometric method in a Warburg apparatus. Wheat roots (150 mg) were placed in vials with the corresponding solutions (3 mL), and after incubation for 10 min respiration was measured every h for 5 h. Results were expressed as μL O_2_ h^−1^ g fresh mass.

### 2.4. H_2_O_2_ Content and Level of Lipid Peroxidation

The H_2_O_2_ content in the roots was determined spectrophotometrically (λ = 560 nm) using the xylenol orange method [36]. The determination of lipid peroxidation was assessed by the accumulation of the thiobarbituric acid (TBA) sensitive products [37].

### 2.5. Gene Expression by Quantitative Real-Time PCR

Total RNA was isolated from wheat roots using the RNeasy Plant Mini Kit (Qiagen, Hilden, Germany) according to the manufacturer’s protocols. RNA concentration and purity were assessed using the NanoDrop^®^ND-1000 spectrophotometer (Thermo Scientific, Wilmington, NC, USA) and the integrity was verified by 1% agarose gel electrophoresis. The reverse transcription (RT) reaction was performed using C1000 Touch™Thermal Cycler (Bio-Rad, Hercules, CA, USA) with the MMLV RT kit (Evrogen, Moscow, Russia). Real-time qPCR was performed using qPCRmix-HS SYBR (Evrogen, Moscow, Russia) and a Real-time CFX Connect (Bio-Rad, Hercules, CA, USA). The templates were amplified three times at 95 °C for 3 min followed by 40 cycles of amplification (94 °C for 10 s and 55/60 °C for 40 s). ADP-ribosylation factor (ARF) and RNase L inhibitor-like protein (RLI) genes were used as reference genes [38]. Sequences of primers for *GAPDH* genes were used from [39]. Primers for *SnRK1* and *ATG* genes are shown in Appendix A.

### 2.6. Statistics

All measurements were taken out in at least 4 biological and 4 analytical replicates. Data are presented as means with standard deviations. Statistical analyses were carried out using the Student’s *t*-test at *p* ≤ 0.05.

## 3. Results

Mitochondria in the control roots excised from the wheat seedlings and incubated in 0.25 mM CaCl_2_ had an orthodox structure with an oval shape, medium density matrix, and numerous weakly expressed cristae (Figure 1a). Changes in mitochondrial morphology were visible after 1 h of exposure of roots to 0.1 μM Mdivi-1. In the presence of Mdivi-1, the size of mitochondria increased approximately 10 times from 0.5 to 1 µm in the control to 5 to 7 µm (Figure 1 and Figure 2). The ultrastructure of mitochondria in Mdivi-1 treated cells was variable. Some mitochondria maintained an orthodox structure with an oval shape (Figure 1b,c), while others became irregularly shaped with membranes forming constriction zones. This may reflect the fusion of the outer and inner membranes of mitochondria and the internal rearrangement of organelles.

Incubation of roots of wheat seedlings in a solution of 0.1 µM Mdivi-1 for 5 h increased O_2_ consumption by approximately 30% (Figure 3). Respiration rates were consistently higher than those of the controls. The mitochondrial potential assessed with the voltage-dependent fluorescent dye TMRM was maintained at a high level during the entire time of exposure of roots to the inhibitor. In the control the membrane potential increased after 3 h and slightly decreased after 5 h (Figure 2).

Analysis of ROS formation by the roots showed that incubation of wheat roots with 0.1 µM Mdivi-1 for 1 h sharply increased H_2_O_2_ content; after 5 h the content decreased to a level lower than that of the control (Figure 4A). In the control, a slight increase in the content of H_2_O_2_ was observed after 3 h of incubation. There were no significant changes in the MDA level both in the control and Mdivi-1 treated roots, indicating that lipid peroxidation had not increased (Figure 4B).

In the cells of the control roots, autophagosomes (visualized as bright LysoTracker stained dots) accumulated after incubation for 3 h in 0.25 mM CaCl_2_; their number decreased after incubation for 5 h (Figure 5). Using this dye results in some nonspecific background staining of cell walls, cytoplasm and perinuclear regions. No autophagosomes were visualized during the entire incubation of the roots with Mdivi-1 (Figure 5). Cell viability remained at control levels during the 24 h of incubation of the roots with Mdivi-1 (data not shown).

Gene expression analysis showed that the level of transcripts of the AMP-dependent protein kinase SnRK1 increased in the control roots after 3 h of incubation and then sharply decreased after incubation for 5 h (Figure 6A). In the presence of Mdivi-1, the expression of *SnRK1* significantly increased after exposure for 5 h. Expression of genes encoding GAPDH isoforms increased after 5 h of exposure to Mdivi-1, while in the control no significant change in the expression of these genes occurred (Figure 6B). The level of transcripts of the autophagic marker protein ATG8 gradually increased in the control during incubation of roots in 0.25 mM CaCl_2_ for 1 to 5 h. While the expression of *ATG8* was also upregulated during incubation of roots with Mdivi-1 for 1 to 5 h, the increase was significantly lower than that of the control (Figure 6C).

## 4. Discussion

The architecture of mitochondria and their dynamic reorganization are regulated by many factors, including dynamin-like proteins belonging to the GTPase family. In the present work, we demonstrated that the application of Mdivi-1, which inhibits dynamin-like protein Drp1, causes significant morphological and physiological consequences in the roots of wheat seedlings. While wounding stress in the control roots is accompanied by increased ROS formation and autophagy, Mdivi-1 treatment promotes mitochondrial fusion, increases energy production, and reduces ROS generation and autophagosome formation. Therefore, mitochondrial fusion appears to mitigate the negative effects of stress.

### 4.1. Mdivi-1 Changes the Ultrastructure and Energy Status of Mitochondria

Changes in mitochondrial morphology and activity (increased respiration and ∆ψH^+^) are often among the first markers of cellular stress [40,41]. In the cells of unstressed plants most mitochondria are small and discrete [42]. In response to stresses such as anoxia, elevated temperature or treatment with mitochondrial poisons, mitochondria can swell or increase in length [19,43,44,45]. Swelling of mitochondria can damage the outer, but not the inner membrane, and trigger the release of cytochrome c [46]. Low temperatures can induce either an increase in the size [47,48] or fragmentation of mitochondria [49]. Thus, careful study of mitochondria structure and function may provide early indicators of plant stress.

Interestingly, wounding stress in wheat roots on its own does not cause dramatic changes in the ultrastructure of mitochondria (Figure 1a), although after application of Mdivi-1 we observed the appearance of large elongated (Figure 1b) or spherical (Figure 1c) mitochondria. The morphology of mitochondrial cristae can be impaired during various stress conditions [50]. However, in our study, the morphology of cristae in large mitochondria was preserved. This indicates that despite causing gross morphological changes, Mdivi-1 did not damage the oxidative phosphorylation of fused mitochondria, because the surface area of cristae positively correlates with the amount of ATP produced by oxidative phosphorylation [50].

The dynamics of mitochondria, including fusion and fission processes, are closely related to their energy status [8]. For mammalian mitochondria, the literature is ambiguous on the effect of Mdivi-1 on enzyme activity of mitochondrial complexes, indicating both inhibition of ETC complex I [51] and an increase in the activity of mitochondrial complexes I, II, and IV [52]. Such differences may be due to the different concentrations of Mdivi-1 used in these studies. Treatment of wheat roots with 0.1 µM Mdivi-1 elevates the energy status of mitochondria, assessed here as increases in mitochondrial respiration rates (Figure 3) and the potential of the mitochondrial membranes (Figure 2). Therefore, fusion of mitochondria induced by Mdivi-1 appears to enhance the functioning of their ETC in wheat roots.

The increased mitochondrial activity that results from fusion requires additional resources, including those supplied by glycolysis. The glycolytic enzyme NAD-dependent glyceraldehyde-3-phosphate dehydrogenase (GAPDH) has a redox-sensitive cysteine in the catalytic center and can therefore reversibly change its own activity, thereby providing a link between redox status and energy metabolism. In Arabidopsis, GAPDH appears to act as an H_2_O_2_ sensor, triggering a protective response to oxidative stress and restoring cellular homeostasis [53]. When cysteine is in the oxidized state, GAPDH can be translocated into the nucleus and mediate signal transfer to induce defense genes [54]. When cysteine is in the reduced state, GAPDH can attach to the outer mitochondrial membrane via the VDAC3 protein, allowing glycolysis to take place next to mitochondria. It is likely that in wheat roots the increased mitochondrial activity induced by Mdivi-1 requires the increased import of glycolytic products such as pyruvate into the mitochondria. Therefore, it is important that the activity of GAPDH is increased, and/or the transcription of corresponding genes is upregulated. This is consistent with the observation in the present study of an upregulation of genes encoding GAPDH after 5 h of exposure of roots to Mdivi-1 (Figure 6A).

Mitochondrial fusion can also influence the key energy sensor SnRK1, the plant orthologue of AMPK (AMP activated protein kinase) in mammalian cells [55]. SnRK1 is an important metabolic regulator involved in stress signaling [56]. In mammalian cells AMPK can phosphorylate the mitochondrial outer membrane receptor MFF, which then binds Drp1, activating mitochondrial fission [57]. The upregulation of *SnRK1* genes found in our experiments (Figure 6B) can reflect metabolic changes and energy shifts that occur in cells in response to Mdivi-1.

### 4.2. Mdivi-1 Treatment Reduces Oxidative Stress

Mitochondria are one of the main sources of ROS in the cells [58]. In our experiments, Mdivi-1 induced a sharp “burst” in H_2_O_2_ content after treatment of roots for 1 h (Figure 2a). This could be due to the activation of the electron transport chain. However, the subsequent reduction in H_2_O_2_ content (Figure 4A) and reduction of lipid peroxidation (Figure 4B) that occurred compared with the controls suggest that temporary rise in H2 fused mitochondria (Figure 3b,c) can effectively cope with oxidative stress. These results are consistent with the decrease in lipid peroxidation observed in Mdivi-1 treated human N2a cells [52]. Thus, by preventing fission GTPase inhibitors cause the appearance of enlarged mitochondria, stimulate ETC functioning and reduce the generation of ROS and associated cell damage.

A consequence of increased levels of ROS is an accumulation of oxidized macromolecules in cells. Previously we showed that the induction of oxidative stress in wheat roots by treating with prooxidant paraquat or antimycin A, an inhibitor of mitochondrial complex III, leads to the intensive formation of autolytic vacuoles (autophagosomes) and the activation of autophagic gene expression [59]. These changes are accompanied by a decrease in the membrane potential of mitochondria and the rate of oxygen consumption. In the present study, autophagosomes were formed in control roots after incubation for 3–5 h in 0.25 mM CaCl_2_ (Figure 5c,e). The initiation of autophagy was confirmed by the upregulation of genes that encode ATG8, a marker protein of autophagy (Figure 6C). ATG8 is a multifunctional ubiquitin-like protein required for the formation of the insulating membrane of autophagosomes, membrane docking, and fusion of the autophagosome with the vacuole/lysosome [60]. The activation of autophagy in the control may have occurred because of the need to degrade cellular structures damaged during the wounding stress caused by the excision of roots from the seedlings. Interestingly, no autophagosomes are observed in the cells of Mdivi-1 treated roots (Figure 5b,d,f), although some upregulation of *ATG8* takes place (Figure 6C). Therefore, in the presence of Mdivi-1 stimulation of mitochondrial activity may reduce the need for autophagy.

## 5. Conclusions

Mitochondrial dynamics is a process that continuously takes place in a cell. The balance between fusion and fission of mitochondria aims to optimize cellular metabolism in response to changes in environmental conditions. In this study, pharmacological inhibition of mitochondrial fission with Mdivi-1 had morphological, physiological, and biochemical effects in *T. aestivum* roots. Enlarged mitochondria, some of irregular shape, were formed, and these mitochondria displayed enhanced functional activity. Upregulation of genes encoding the glycolytic enzyme GAPDH and the AMP-dependent protein kinase SnRK1 suggests that redistribution of energy-producing metabolic pathways occurs to sustain the supply of substrates to fused mitochondria. Controlling mitochondrial fusion rates could fundamentally change the physiology of wheat plants by altering the energy status of the cell. Compared to the untreated wounded control, the inhibition of mitochondrial fission had the effects of decreasing ROS formation and associated cell damage. Therefore, the inhibition of fission and resulting fusion of mitochondria might mitigate negative stress effects in wheat roots. In a broader context, mitochondrial fusion and fission events may provide important adaptive responses to environmental stresses, and the ability to control these processes could provide targets for the future breeding and genetic modification of crops.

## Figures and Tables

**Figure 1 life-12-01386-f001:**
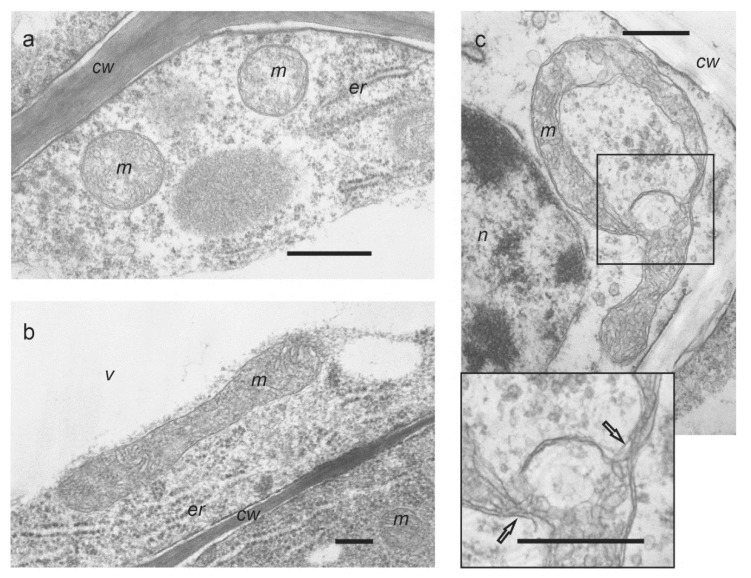
Ultrastructure of mitochondria in wheat roots: (**a**) orthodox mitochondria of oval shape in control roots (1 h); (**b**) enlarged mitochondrium in roots treated with 0.1 μM Mdivi-1 (1 h); (**c**) mitochondrium of irregular shape in Mdivi-1 treated roots (1 h). In the increased fragment, the arrows indicate constriction zones. *cw*—cell wall, *er*—endoplasmic reticulum, *m*—mitochondrium, *n*—nucleus, *v*—vacuole. Scale bar is 0.5 μm.

**Figure 2 life-12-01386-f002:**
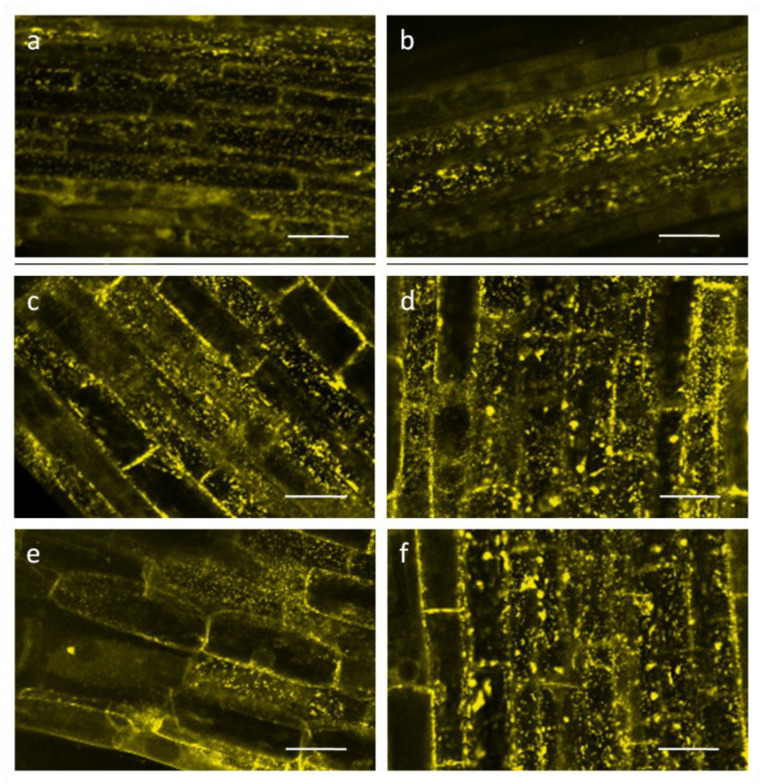
Mitochondrial membrane potential visualized with TMRM: (**a**,**c**,**e**)—control for 1, 3, 5 h, respectively; (**b**,**d**,**f**)—roots treated with 0.1 μM Mdivi-1 for 1, 3, 5 h, respectively. Scale bar is 50 µm.

**Figure 3 life-12-01386-f003:**
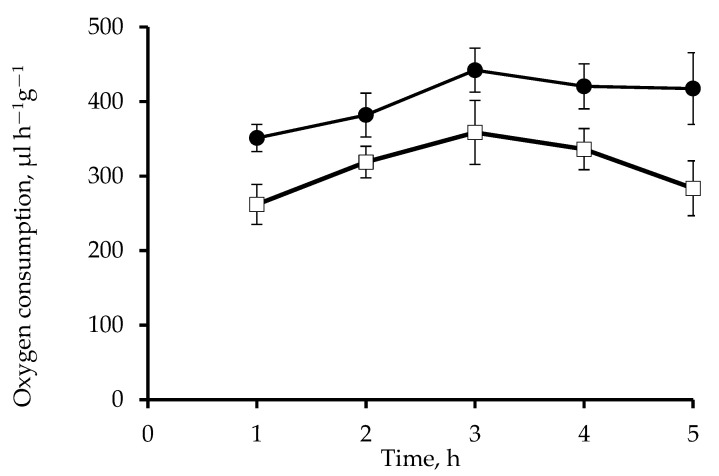
Oxygen consumption of wheat roots treated with 0.1 μM Mdivi-1. Control—white squares, Mdivi-1—black circles. Treatment with Mdivi-1 had a significant effect, *p* ˂ 0.05.

**Figure 4 life-12-01386-f004:**
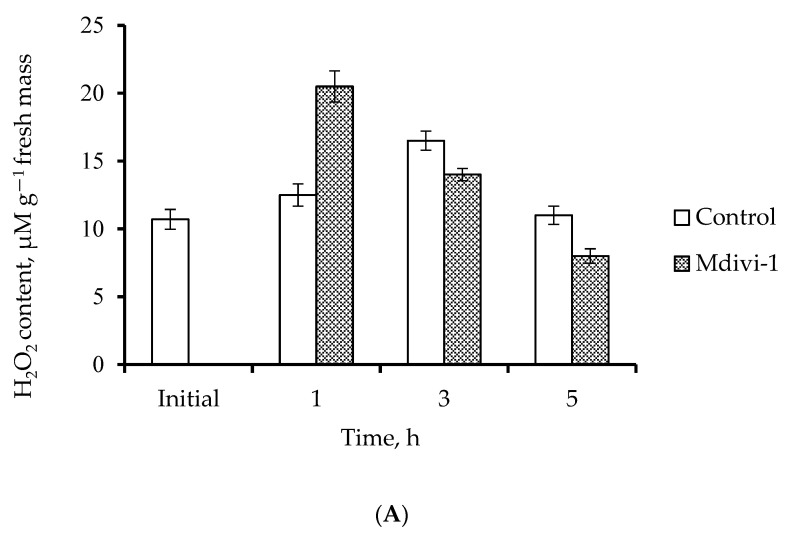
Content of H_2_O_2_ (**A**) and malondialdehyde (MDA) (**B**) in wheat roots following treatment with 0.1 μM Mdivi-1.

**Figure 5 life-12-01386-f005:**
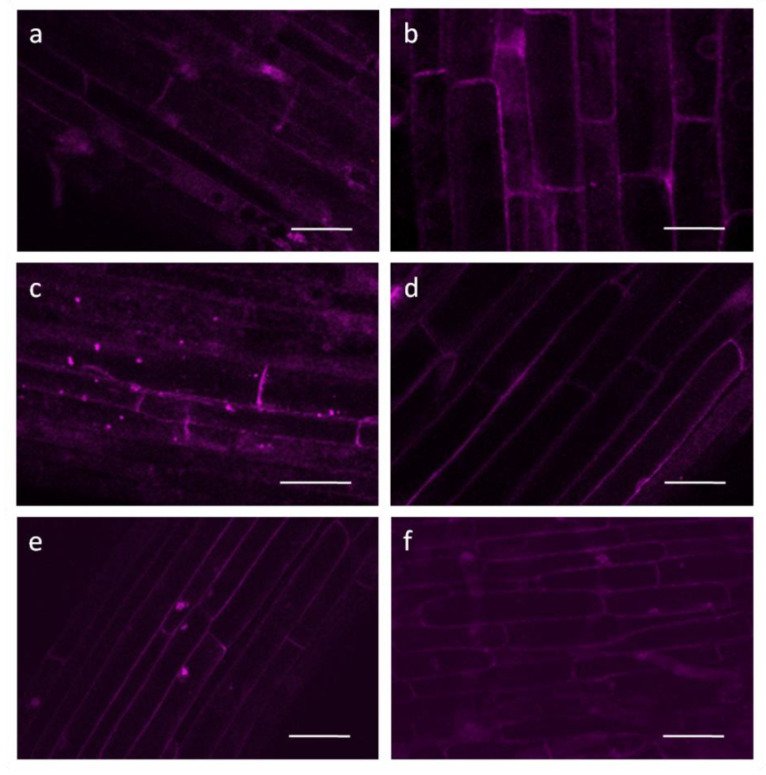
Autophagosome formation visualized with LysoTracker Red: (**a**,**c**,**e**)—control for 1, 3, 5 h, respectively; (**b**,**d**,**f**)—roots treated with 0.1 μM Mdivi-1 for 1, 3, 5 h, respectively. Scale bar is 50 µm.

**Figure 6 life-12-01386-f006:**
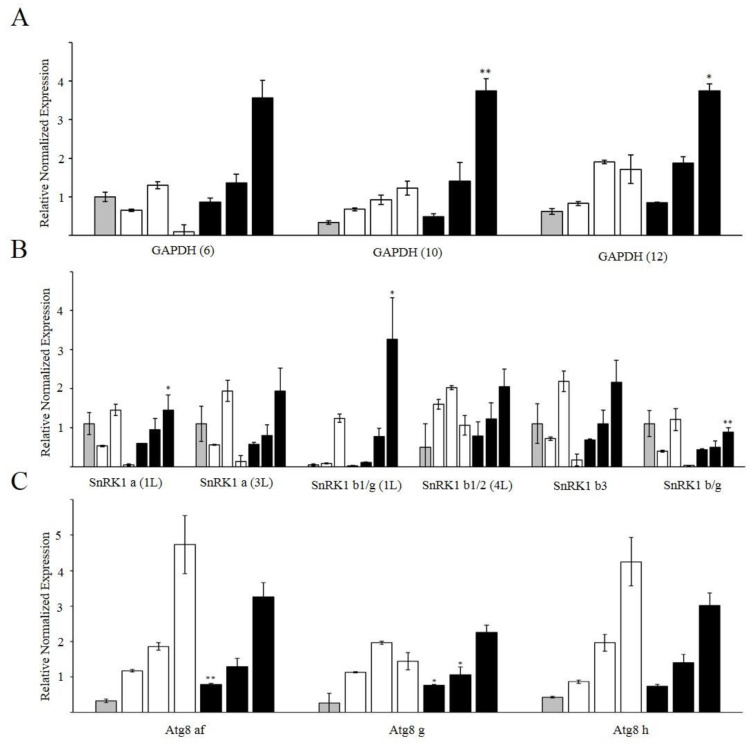
Gene expression of wheat roots cells treated with 0.1 μM Mdivi-1: (**A**)—isoforms of glyceraldehyde-3-phosphate dehydrogenase (GAPDH); (**B**)—subunits of sucrose non-fermenting-related kinase (SnRK1); (**C**)—isoforms of ATG8 genes. Grey bar—intact roots, white bars—control roots for 1, 3, 5 h, black bars—roots treated with 0.1 μM Mdivi-1 for 1, 3, 5 h. * *p* ≤ 0.05; ** *p* ≤ 0.005.

## Data Availability

Not applicable.

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
