# Peer review of "Mdivi-1 Induced Mitochondrial Fusion as a Potential Mechanism to Enhance Stress Tolerance in Wheat"

_life, 2022, doi:10.3390/life12091386_

Round 1

Reviewer 1 Report

The authors investigated the role of mitochondrial fusion in wheat root cells, under wounding stress simulated by excision, based on the inhibitor of the mitochondrial fragmentation protein Drp1 (Mdivi-1). The size of the mitochondria was increased to enhance their functional activity and elevate their membrane potentials, while a decrease in ROS formation and associated cell damage was observed. More importantly, the mitochondrial fusion was associated with a general activation of energy metabolism.

I believe that the authors have provided sufficient background, explained well the methodologies used, presented the data using appropriate tables and figures, and concluded appropriately based on available data. I also believe that the authors have explored the issues of the plant growth under stresses from a quite unique perspective. I have one major concern and several minor suggestions that I would like to provide for the authors to consider if a revision is requested by the editor.

Major issue:

I have hard time understanding the statements that the authors claimed as “In a broader context, the ability to control mitochondrial fusion and fission could provide targets for future breeding and genetic modification of crops” in Abstract. I wish that the authors could elaborate this conclusion somewhere in the main text.

Minor issues:

--why the wheat is selected for this study?

--line 102, the “Abrabidopsis” needs to be italicized

--line 113, the abbreviation should follow the full name

--appropriate spacing is needed in both sections of Results and Discussion

--line 257, this subsection title is out of space, needs to be removed

--Table S1 should not be included in the main text

Author Response

The authors investigated the role of mitochondrial fusion in wheat root cells, under wounding stress simulated by excision, based on the inhibitor of the mitochondrial fragmentation protein Drp1 (Mdivi-1). The size of the mitochondria was increased to enhance their functional activity and elevate their membrane potentials, while a decrease in ROS formation and associated cell damage was observed. More importantly, the mitochondrial fusion was associated with a general activation of energy metabolism.

I believe that the authors have provided sufficient background, explained well the methodologies used, presented the data using appropriate tables and figures, and concluded appropriately based on available data. I also believe that the authors have explored the issues of the plant growth under stresses from a quite unique perspective.

We thank a reviewer for high opinion of our research.

I have one major concern and several minor suggestions that I would like to provide for the authors to consider if a revision is requested by the editor.

Major issue:

I have hard time understanding the statements that the authors claimed as “In a broader context, the ability to control mitochondrial fusion and fission could provide targets for future breeding and genetic modification of crops” in Abstract. I wish that the authors could elaborate this conclusion somewhere in the main text.

We agree that this claim is a bit speculative and therefore deleted it from the Abstract.

Minor issues:

--why the wheat is selected for this study?

This manuscript is submitted to the Special issue dedicated to wheat. Wheat is a staple food throughout the world, an important source of carbohydrates and proteins. Therefore, to study stress tolerance mechanisms in this crop is vitally important.

--line 102, the “Abrabidopsis” needs to be italicized - done

--line 113, the abbreviation should follow the full name - done

--appropriate spacing is needed in both sections of Results and Discussion

Not sure which spacing referee means

--line 257, this subsection title is out of space, needs to be removed

--Table S1 should not be included in the main text

These minor issues are corrected in the revised version.

Reviewer 2 Report

life-1906648-peer-review-v1

Article:  Mdivi-1 induced mitochondrial fusion as a potential mechanism to enhance stress tolerance in wheat.

The idea of this study is distinctive and interesting; it studied the link between the mitochondrial morphology from one side and the environmental stress and energy status from the other side. As well as, this study clarified the importance of the mitochondrial fusion in its functional activity inside plant cell (root of wheat plants)

The quality of TEM sections is satisfied and can obviously reflect the hypothesis of authors, in general, the manuscript is well written in all its sections. The manuscript needs minor improvements:

·         First, there is no need for the simple summary, to repeat in the abstract.

·         In the introduction part we can focus more on Mdivi-1.

·         kiss-and-run in line 56, please delete

·         Line 116: Add the author's name (32).

·         Line 129-133: The paragraph is redundant in the introduction, can be moved to the results section.

·         Please explain the method of sowing the seeds in some detail, as well as the age of the seedling.

·         DMSO ??

·         Line 138: Authors mentioned "Roots were excised from seedlings and incubated in 0.25 mM CaCl2 138 (control)" ,,, Please state the reason for choosing CaCl2 and that concentration with documentation with reference.

·         What is the source of Mdivi-1?

·         Supplementary Materials: The following supporting information can be downloaded at: 453 www.mdpi.com/xxx/s1, Table S1: Primers used for qPCR. (The site did not open).

Author Response

Article: Mdivi-1 induced mitochondrial fusion as a potential mechanism to enhance stress tolerance in wheat.

The idea of this study is distinctive and interesting; it studied the link between the mitochondrial morphology from one side and the environmental stress and energy status from the other side. As well as, this study clarified the importance of the mitochondrial fusion in its functional activity inside plant cell (root of wheat plants)

The quality of TEM sections is satisfied and can obviously reflect the hypothesis of authors, in general, the manuscript is well written in all its sections.

We thank a reviewer for high opinion of our work.

The manuscript needs minor improvements:

  • First, there is no need for the simple summary, to repeat in the abstract.

“Simple summary” is an obligatory section, which is a part of submitted manuscript. We shortened it to avoid repetition with the Abstract.

  • In the introduction part we can focus more on Mdivi-1.

Unfortunately, very little information in the literature is available about the effects of Mdivi-1 on plant physiology. Previously and in present work we used this compound as a tool to simulate mitochondrial fusion in root cells and to discuss physiological consequences of fused mitochondria for wheat plants in stress conditions.

  • kiss-and-run in line 56, please delete
  • Line 116: Add the author's name (32).

These minor issues are corrected in the revised version.

  • Line 129-133: The paragraph is redundant in the introduction, can be moved to the results section.

We think this is optional and a very brief mention of our main experimental findings without detailed description, like in the Results section, is logical here.

  • Please explain the method of sowing the seeds in some detail, as well as the age of the seedling.

Wheat (Triticum aestivum L. cv. Kazanskaya Jubilejnaya) seeds were distributed evenly in a single layer and sown on a tissue wetted with distilled water on a sheet of glass for 24 h. After that imbibed seeds were transferred to the containers with 0.25 mM CaCl2 solution and seedlings were grown hydroponically for 5 d at 25 °C with a 12/12 photoperiod. Text was added in the Materials and Methods.

  • DMSO ??

Dimethylsulfoxide

  • Line 138: Authors mentioned "Roots were excised from seedlings and incubated in 0.25 mM CaCl2 138 (control)" ,,, Please state the reason for choosing CaCl2 and that concentration with documentation with reference.

CaCl2 at low (0.25 mM) concentration was used in our experiments as a membrane stabilizing compound in wheat seedlings. Previously we showed that the performance of plants depends on the extent to which they can maintain membrane fluidity and prevent lipid phase transition in response to stresses. This was supported by the analysis of lipid composition and the index of membrane stability in the roots of wheat seedlings grown on 0.25 CaCl2 solution.

Valitova Yu.N.; Khabibrakhmanova V.R.; Belkina A.V.; Renkova A.G.; Minibayeva F.V. Lipid profile of wheat roots under the action of membranotropic agents. Biologicheskie Membrany 2020, 37, 466-476. doi 10.31857/S0233475520060080

  • What is the source of Mdivi-1?

Sigma-Aldrich, China